# A Sub-Bottom Type Adaption-Based Empirical Approach for Coastal Bathymetry Mapping Using Multispectral Satellite Imagery

Xue Ji [1,2,3], Yi Ma [1,4], Jingyu Zhang [1,4,*], Wenxue Xu [4] and Yanhong Wang [4]

1 Technology Innovation Center for Ocean Telemetry, Ministry of Natural Resources, Qingdao 266061, China; jixuesdqd@jlu.edu.cn (X.J.); mayimail@fio.org.cn (Y.M.)
2 State Key Laboratory of Information Engineering in Surveying, Mapping and Remote Sensing, Wuhan 430079, China
3 College of Geoexploration Science and Technology, Jilin University, Changchun 130026, China
4 First Institute of Oceanology, Ministry of Natural Resources, Qingdao 266061, China; xuwx@fio.org.cn (W.X.); wangyanhong@fio.org.cn (Y.W.)
* Correspondence: zhangjingyu@fio.org.cn

**Abstract:** Accurate bathymetric data in shallow water is of increasing importance for navigation safety, coastal management, and marine transportation. Satellite-derived bathymetry (SDB) is widely accepted as an effective alternative to conventional acoustic measurements in coastal areas, providing high spatial and temporal resolution combined with extensive repetitive coverage. Many previous empirical SDB approaches are unsuitable for precision bathymetry mapping in various scenarios, due to the assumption of homogeneous bottom over the whole region, as well as the neglect of various interfering factors (e.g., turbidity) causing radiation attenuation. Therefore, this study proposes a bottom-type adaption-based SDB approach (BA-SDB). Under the consideration of multiple factors including suspended particulates and phytoplankton, it uses a particle swarm optimization improved LightGBM algorithm (PSO-LightGBM) to derive depth of each pre-segmented bottom type. Based on multispectral images of high spatial resolution and in situ observations of airborne laser bathymetry and multi-beam echo sounder, the proposed approach is applied in shallow water around Yuanzhi Island, and achieves the highest accuracy with an RMSE value of 0.85 m compared to log-ratio, multi-band, and classical machine learning methods. The results of this study show that the introduction of water-environment parameters improves the performance of the machine learning model for bathymetric mapping.

**Keywords:** satellite-derived bathymetry; airborne laser bathymetry; seafloor substrates; coastal bathymetry mapping

## 1. Introduction

Bathymetry in shallow coastal regions plays a decisive role in reconnaissance surveys, marine spatial planning (MSP), urban development as well as scientific research, such as the understanding of erosion dynamics and evolution, and estimates of fixed quantities for terrestrially-derived carbon [1,2]. A range of modern techniques have been employed in bathymetry determination benefited by the continuous development of Geo-Information systems and remote sensing technology [3–5]. Among them, vessel-based acoustic sonars (single- and multi-beam) are conventionally used for hydrographic surveying [6,7]. However, due to the inherent difficulties arising from the intricate and dynamic characteristics, e.g., the presence of submerged obstacles and irregular bottom topography, vessel-based approaches are often prohibitive and disappointing when dealing with complex areas such as littoral zones and massive hidden reefs with water depth less than 15 m. Airborne laser bathymetry (ALB), developed rapidly in recent decades, is praised as the gold standard of

coastal mapping, and able to generate high-resolution and accurate bottom topography range from 0.5 m to 70 m over clear waters [8,9]. Such a method is generally logistically unfeasible to adapt to semi-turbid and turbid waters due to the limitation of optical signal transmitting in the water column. Neither shipborne nor airborne sensors are operable for producing accurate large-scale bathymetry, as they are labor-intensive, prohibitively expensive, and time-consuming, restricting repetitive and cost-effective data acquisition [10]. Furthermore, synthetic aperture radar (SAR), an active microwave remote sensing system, is able to achieve all-weather, all-day earth observation without being subject to cloud cover. However, SAR has not been widely used in marine engineering due to its sensitivity to wind and low accuracy of the derived depth [11,12]. By comparison, satellite-derived bathymetry (SDB), another remote sensing tool, offers a more flexible, repeatable, efficient, and cost-effective means to map coastal bathymetry [13–15].

A number of methods are employed to assess SDB [16–19], which fall into two broad categories: statistical-based approaches and physics-based approaches [11]. Historically, statistical-based approaches can be further subdivided into analytical, semi-analytical, quasi-analytical, and semi-empirical methods [20]. In general, physics-based approaches are used to obtain bathymetric information through inherent optical properties (IOPs)-based radiative transfer model or its simplified version, as well as forward modeling algorithms under the assumption of a fixed substrate and water quality [21,22]. The widely applied physics-based models, including the flow radiative transfer model [23] and its variation [2,24], possess satisfactory depth evaluation accuracy and physical universality independent of ground-truthed data. However, intricate optical properties of the water column are usually essential in the construction of these models, including but not limited to the spectral characteristics of suspended solids and solutes and bottom reflectance, such as chlorophyll-*a* (Chl-*a*) concentration, diffuse attenuation coefficient of the water body, detritus concentration, spectral shape, absorption, and backscattering coefficient. Subsequently, the physics-based models are further simplified and adjoined with empirical parameters to accomplish bathymetry mapping, and a variety of semi-analytical and semi-empirical models are developed, such as single-band [25], dual-band [19], multi-band generalized linear [26,27], and log-ratio method (LRM) [3,13,28–30]. Apart from the fuzziness of a large number of estimated parameters, the limitation of assumptions is considered as the major factor restricting the performance and accuracy of models, assuming that throughout the study area (1) the attenuation of a beam in the water column follows the Beer law, which is an exponential function of bathymetry; (2) water turbidity is uniform, and it is mainly assumed as Case 1 water; (3) the bottom cover reflection characteristic is homogeneous; (4) atmospheric and wave conditions are uniform and similar [18,21,22]. Compared with the physics-based method, which constructs a theoretical framework or general strategy based on the radiative transfer model of water, empirical methods drive advanced statistical approaches or regression approaches to confirm the mapping relationship between spectral qualities and depth, without considering the IOPs and water column properties, and have become the research emphasis in recent decades [10,31,32].

These empirical methods have been accepted and integrated as a survey tool by an increasing number of researchers [20]. Recently, machine learning (ML) has been proven to be able to offer more satisfactory solutions such as random forest (RF) [14,31], support vector machine (SVM) [14,33–35], bagging, least squares boosting (LSB), K-nearest neighbor (KNN) [36], back propagation neural network (BPNN) [37], recursive neural network (RNN), radial basis function (RBF) [38], and deep learning model (DP) [39–42]. However, the consideration factors of major models are limited to the blue-green band reflectivity or visible spectrum, ignoring the water column properties as well as the nature of the seafloor that influence the inversion quality of water depth. In addition, the deep-water (more than 10 m) accuracy is sacrificed in seeking the optimal global solution. In conclusion, to identify the limitations of most current models, which are preferentially applied in shallow-water regions with specific water conditions and uniform substrate

types, rather than being adaptive to different sea areas with various water qualities or substrate types, we attempt to propose a new empirical SDB approach. It possesses the following advantages: (1) adaptive segmentation of different substrates and construction of corresponding mapping relationship; (2) in addition to the visible spectrum, main factors affecting water-leaving radiance and the characteristics with specific mapping ability for water depth are included in the discussion and adaptively screened based on different substrate types; (3) the advanced regression analysis model LightGBM is used for SDB, and the optimal parameters can be adaptively adjusted based on a particle swarm optimization (PSO) algorithm to improve the accuracy of bathymetry derivation.

## 2. Materials and Methods

### 2.1. Study Site

We applied and tested our new approach to Yuanzhi Island, located in the western part of the Yongle Atoll (Figure 1). The island of Yuanzhi has an approximate area of 0.3 km$^2$ and an elliptical shape, spanning ~700 m in the north-south direction and ~500 m in the east-west direction. It is one of the few islands with fresh water in the Xisha Islands, making it of great research value.

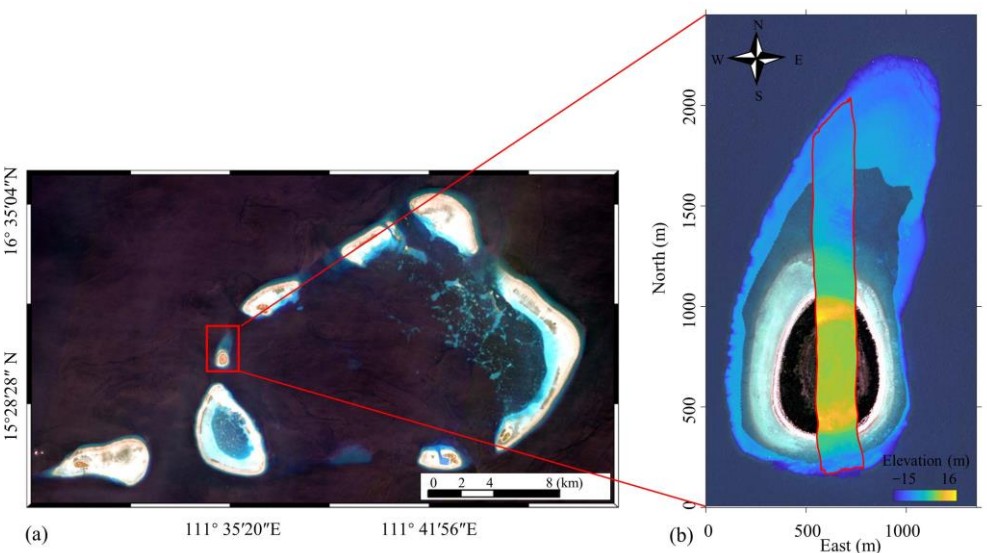

**Figure 1.** Locations of our study sites. (**b**) Water depth at the selected area in the box in (**a**), where the area bounded by red is the water depth obtained by ALB, and the rest is the water depth obtained by multi-beam sonar.

### 2.2. Methodology

#### 2.2.1. Overview

Figure 2 shows that the radiance received by the detector is limited by various confounding factors in solar illumination. Solar illumination is subjected to absorption and scattering from the atmosphere, water-body, and seafloor substrate. The principle states that total radiative energy reflected by the receiver is a function of atmosphere, bottom reflectance, water depth, and water column properties related to water turbidity, such as Chl-*a* concentration, diffuse attenuation coefficient, suspended organic matter, reflectance of the water column, as well as suspended sediment [16,43,44]. Obviously, it is necessary to construct the corresponding relationship between water depth and reflectivity considering the main influential factors, and it is theoretically imperfect to assume that coefficients for the water column or bottom reflectivity spectra remain constant over the target scene.

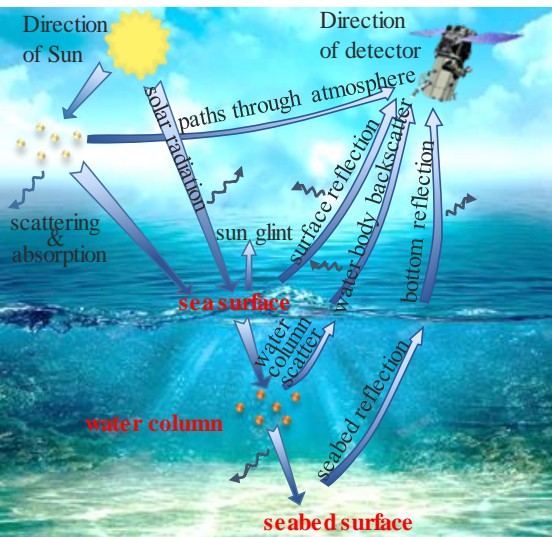

**Figure 2.** The various confounding factors that diminish radiance across the atmosphere, water mass, and substrate. The wavy arrows refer to scattering and absorption during the propagation of solar illumination.

This study attempts to determine the bathymetry over the study area with complex and unknown substrate through a bottom-type adaption-based SDB approach (BA-SDB) that possesses flexible adaptability and robust performance. The working draft of BA-SDB is schematically shown in Figure 3.

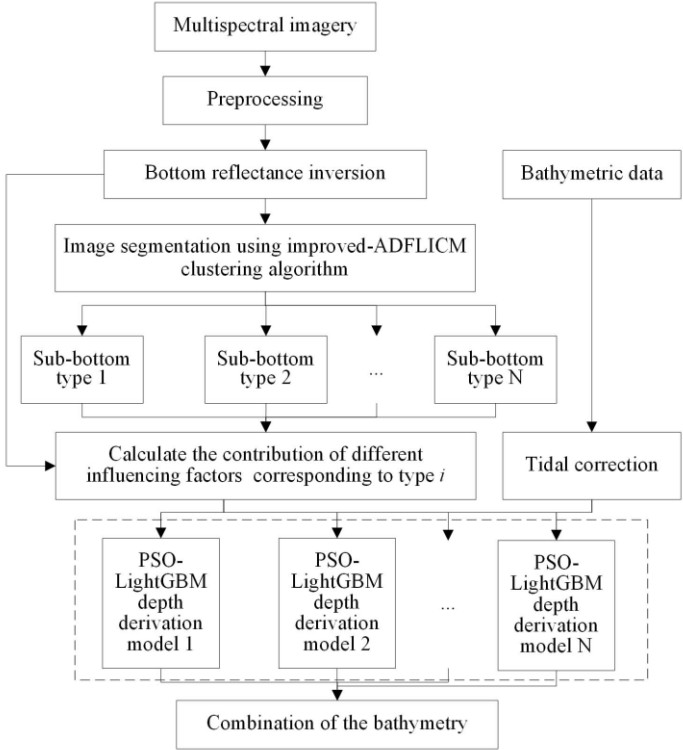

**Figure 3.** Working draft to derive bathymetry driving the bottom-type self-adaption-based SDB model.

### 2.2.2. Satellite Images and Field Survey Data

The WorldView-2 (WV-2) image collected by Maxar Digital GlobeTM on November 18, 2014, provides radiance in 8 wavelengths (coastal, blue, green, yellow, red, red edge,

NIR-1, and NIR-2) ranging from the visible to the near-infrared spectrum, with a spatial resolution of 1.8 m.

Bathymetry data were collected by both multibeam echosounder (MBES) and airborne laser bathymetry system (ALB). The data of ALB were acquired utilizing the Optech Aquarius system in 2013. The ALB system employed in this study uses a pulsed Nd:YAG laser head operating at a frequency of 70 kHz to generate a doubled green beam with a wavelength of 532 nm. It possesses a root mean square error (RMSE) of 0.25 m for depth measurements. The divergence of the laser beam was specified as 1 mrad, with a pulse width of 8.3 ns. The scanning nadir angle of the laser beam was set at 20°, while the detection frequency was set to 550 kHz. A point density ranging from 5–10 points per square meter was achieved through the measurement process. The MBES employed in this study is the SONIC 2024 system. The system achieves a span resolution of 1.25 cm, covers a width ranging from 10° to 160°, and encompasses 256 beams for data acquisition. The resulting point density for measurements obtained is within the range of 10–20 points per square meter. However, due to the extreme shallowness of the water and safety concerns, which prohibited the boat from approaching the survey area, no MBES data were available for the nearshore region of Yuanzhi Island.

### 2.2.3. Preprocessing

Image processing consists of refined geo-referencing through measured DEM, radiometric calibration, de-lighting, atmospheric correction, and land–water segmentation to standardize all WV-2 images for precision comparison and analysis.

WV-2 image processing includes radiometric and geometric correction, utilizing processing software or programming to generate subsurface remote sensing reflectance. Subsequently, the Hedley model [45] was applied to rectify glinted pixels and normalize all images. Figure 4a,b illustrate the RGB images prior to and after preprocessing, respectively. Two detailed views in Figure 4 further illustrate the effectiveness of the process for sun glint caused by specular reflection. Image division of land, invalid waters, and effective waters is completed using NDWI coefficients, blue and green band reflectance. The results of this division are then superimposed in Figure 4, with the blue closed curve representing the land area and the red curve corresponding to the invalid water area.

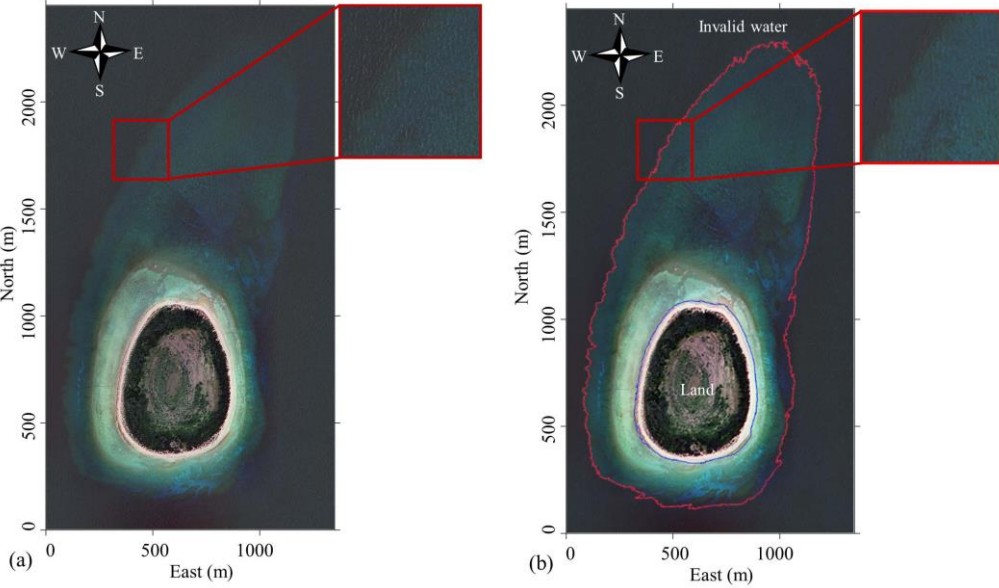

**Figure 4.** Comparison of satellite imagery before and after preprocessing. (**a**) The original WV-2 image and (**b**) the pre-processed image.

### 2.2.4. Inversion of Bottom Reflectance

Most empirical models for bottom reflectance inversion rely on complex pre-parameters such as seafloor type, diffuse attenuation coefficient, bathymetry, etc. [46]. Given all this, Ma et al. constructed an exponential algorithm for bottom reflectance inversion, the LR-S model, with no need for ground data. On the basis of the log-ratio model proposed by Stumpf [30], the LR-S model randomly selects each self-inference point on the corrected images through spectral features to obtain its bottom reflectance [47]. In the LR-S model, the bottom reflectance at 550 nm is defined as:

$$R_{bottom} = \alpha_1 \times exp\left(\alpha_2 \times C_i'\right) \tag{1}$$

where $C_i'$ is the logarithmic value of the blue- and green-band subsurface reflectance after rotation, $r_{rs}(\text{Blue})$ and $r_{rs}(\text{Green})$ correspond to subsurface reflectance after rotation in blue and green bands, respectively, and $\alpha_1$ and $\alpha_2$ are parameters in the inversion function. For a detailed description, please refer to the paper by Ma et al. [47].

### 2.2.5. Adaptive Bottom Substrate Partitioning

As a significant imaging factor in the SDB study, the bottom type needs to be considered. The ADFLICM method proposed by Zhang et al. is a new adaptive fuzzy local information *c*-means clustering method, which enables the weight factor of the adjacent pixel effect to adaptively determine and enhance the stability of the patchy homogeneous classification, while reducing the edge blurring effect by merging local space and gray level information constraints [48]. $R_{bottom}$ is used instead of the grayscale image stretched to the [0, 255] range as input for clustering.

Based on the novel local similarity measure $S_{ir}$, the objective function $J_m$ of the AD-FLICM method is described as follows:

$$J_m = \sum_{i=1}^{N} \sum_{k=1}^{C} u_{ki}^m \left[ \|p_i - v_k\|^2 + \frac{1}{N_R} \sum_{r \in N_i} (1 - S_{ir}) \|p_r - v_k\|^2 \right], \tag{2}$$

$$v_k = \frac{\sum_{i=1}^{N} u_{ki}^m \left( p_i + \frac{1}{N_R} \sum_{r \in N_i} (1 - S_{ir}) \times p_r \right)}{\sum_{i=1}^{N} u_{ki}^m \left( 1 + \frac{1}{N_R} \sum_{r \in N_i} (1 - S_{ir}) \right)}. \tag{3}$$

where $p_i$ denotes the reflectance of the *i*th pixel, $v_k$ is the prototype value of the *k*th cluster, and $u_{ki}$ is the degree of fuzzy membership of $p_i$ belonging to the *k*th cluster. In addition,

$$u_{ki} = \sum_{j=1}^{C} \left( \frac{\|p_i - v_k\|^2 + \frac{1}{N_R} \sum_{r \in N_i} (1 - S_{ir}) \|p_r - v_k\|^2}{\|p_i - v_j\|^2 + \frac{1}{N_R} \sum_{r \in N_i} (1 - S_{ir}) \|p_r - v_j\|^2} \right)^{-\frac{1}{m-1}}, \tag{4}$$

where *m* is the weight exponent for each fuzzy membership, $N_R$ represents its cardinality, $N_i$ denotes the set of neighborhood pixels in the window ($0 < (x_i - x_r)^2 + (y_i - y_r)^2 \leq 2L - 1$), where $(x_i, y_i)$ and $(x_r, y_r)$ are the coordinates of pixels *i* and *r*, respectively, and *L* represents the level of the neighborhood) around $p_i$, and $p_r$ is the neighborhood pixel that falls into $N_i$.

To further propel the adaptive determination of clustering parameters, we improve the ADFLICM method by introducing the Calinski–Harabasz index to the original algorithm framework [49]. The Calinski–Harabasz Index $\text{CHi}(k)$ [49] is used to calculate the optimal clustering number within a specific range ($2 < k$). Normalized reflectance before evaluation. For a randomly selected set of data *E* of size $N_E$ after clustering, the $\text{CHi}(k)$ is defined by:

$$\text{CHi}(k) = \frac{\text{tr}(B_k)}{\text{tr}(W_k)} \times \frac{n_g - k}{k - 1}, \tag{5}$$

i.e., the ratio of the mean value of inter-cluster dispersion to the mean value of intra-cluster dispersion. In the formula, $\text{tr}(B_k)$ is the trace of the between-group dispersion matrix

and $tr(W_k)$ corresponds to the trace of the within-cluster dispersion matrix, which are defined as:

$$W_k = \sum_{q=1}^{k} \sum_{p \in C_q} (p - c_q)(p - c_q)^T, \tag{6}$$

$$B_k = \sum_{q=1}^{k} n_q (c_q - c_E)(c_q - c_E)^T. \tag{7}$$

For cluster $q$, the corresponding point set is $C_q$ with point number $n_q$, and the center is $c_q$. Similarly, $c_E$ is the center of $E$. Theoretically, the higher the value of CHi, the more optimal the number of clusters.

### 2.2.6. Bathymetry Algorithm

Constructing a regression model and taking into account water depth factors are essential for optimizing the performance of empirical modeling. SDB approaches are increasingly focusing on water turbidity and sub-bottom benthos or vegetation within the limits of optical-derived bathymetry. This is because these factors have been found to affect the accuracy of bathymetric maps created using remote sensing techniques. To create a derived model for each substrate type that accounts for water turbidity, suspended sediment concentration (SSC) and chlorophyll-*a* concentration were included in the regression model, together with the reflectance of each band. Table 1 lists the water depth factors fed into the regression model.

**Table 1.** Statistical table of water depth factors.

| No. | Description | Equation | No. | Description | Equation |
|---|---|---|---|---|---|
| 1 | Suspended sediment factor | $\frac{r_{rs}(\text{Green}) + r_{rs}(\text{Red})}{\frac{r_{rs}(\text{Blue})}{r_{rs}(\text{Green})}}$ | 2 | chlorophyll-*a* concentration | $\text{Chl}\_a = 10^{(-0.4909 + 191.659 * w)}, w = r_{rs}(\text{Green}) - 0.46 * r_{rs}(\text{Red}) - 0.54 * r_{rs}(\text{Coastal})$ |
| 3 | Coastal band reflectance | $r_{rs}(\text{Coastal})$ | 4 | Blue band reflectance | $r_{rs}(\text{Blue})$ |
| 5 | Green band reflectance | $r_{rs}(\text{Green})$ | 6 | Yellow band reflectance | $r_{rs}(\text{Yellow})$ |
| 7 | Red band reflectance | $r_{rs}(\text{Red})$ | 8 | Red edge band reflectance | $r_{rs}(\text{Red edge})$ |
| 9 | NIR-1 band reflectance | $r_{rs}(\text{NIR-1})$ | 10 | NIR-2 band reflectance | $r_{rs}(\text{NIR-2})$ |
| 11 | ratio of $r_{rs}(\text{Coastal})$ to $r_{rs}(\text{Blue})$ | $\frac{r_{rs}(\text{Coastal})}{r_{rs}(\text{Blue})}$ | 12 | ratio of $r_{rs}(\text{Coastal})$ to $r_{rs}(\text{Green})$ | $\frac{r_{rs}(\text{Coastal})}{r_{rs}(\text{Green})}$ |
| 13 | ratio of $r_{rs}(\text{Coastal})$ to $r_{rs}(\text{Yellow})$ | $\frac{r_{rs}(\text{Coastal})}{r_{rs}(\text{Yellow})}$ | 14 | ratio of $r_{rs}(\text{Blue})$ to $r_{rs}(\text{Green})$ | $\frac{r_{rs}(\text{Blue})}{r_{rs}(\text{Green})}$ |
| 15 | ratio of $r_{rs}(\text{Blue})$ to $r_{rs}(\text{Yellow})$ | $\frac{r_{rs}(\text{Blue})}{r_{rs}(\text{Yellow})}$ | 16 | ratio of $r_{rs}(\text{Green})$ to $r_{rs}(\text{Yellow})$ | $\frac{r_{rs}(\text{Green})}{r_{rs}(\text{Yellow})}$ |
| 17 | log-ratio of $r_{rs}(\text{Coastal})$ to $r_{rs}(\text{Green})$ | $\ln \frac{r_{rs}(\text{Coastal})}{r_{rs}(\text{Green})}$ | 18 | log-ratio of $r_{rs}(\text{Blue})$ to $r_{rs}(\text{Green})$ | $\ln \frac{r_{rs}(\text{Blue})}{r_{rs}(\text{Green})}$ |

Before deriving, it is necessary to calculate Spearman's correlation coefficient $\rho$ for each reflectance of different bands and their combinations $x$ and water depth $d$. For each type of substrate, the characteristics corresponding to the ten principal correlation coefficients are entered into the regression model. Here, $\rho$ is obtained by Equation (8). $x$ and $d$ represent the mean of $x$ and $d$, respectively.

$$\rho = \frac{\sum_i (x_i - x)(d_i - d)}{\sqrt{\sum_i (x_i - x)^2 \sum_i (d_i - d)^2}} \tag{8}$$

The LightGBM regression model [50], proposed by a Microsoft team in 2017, is an ensemble algorithm based on a gradient-boosting decision tree (GBDT) that improves the regression accuracy and arithmetic speed with the unique gradient-based one-side sampling (GOSS) strategy and exclusive feature bundling (EFB) strategy. It is optimized by a particle swarm optimization (PSO) algorithm [51] to obtain accurate derived bathymetry. Under the GOSS strategy, all samples with more significant gradients are retained, while

the remaining gradient samples are randomly sampled. This allows models with significant training errors to get more attention without changing the data distribution.

The decision tree is a form of supervised learning that learns the function from the feature vector $x_i$ to the gradient $g_i$, and splits each node at the maximum information gain. Assume that $O$ is the train data on the fixed node of the decision tree, and the variance gain $V$ of split feature $j$ at point $d$ is defined as:

$$V_{j|O}(d) = \frac{1}{n_O} \left( \frac{\left( \sum_{\langle x_i \in O : x_{ij} \leq d \rangle} g_i \right)^2}{n_{l|O}^j(d)} + \frac{\left( \sum_{\langle x_i \in O : x_{ij} > d \rangle} g_i \right)^2}{n_{r|O}^j(d)} \right). \tag{9}$$

In the formula, $n_O$ represents the total number of samples on the node, $n_{l|O}^j(d)$ denotes the number of samples on the node less than the split point $d$, $n_{r|O}^j(d)$ is the sample quantity on the node greater than $d$. The optimal split point corresponding to feature $j$ is $d_j^*$, and the maximum gain $V_j\left(d_j^*\right)$ is then calculated.

Subset $A$ consists of the top $a \times 100\%$ of the data sorted in descending order of the absolute value of the gradient, while subset $A^c$ consists of the remaining data. Subset $B$ is formed by randomly sampling $b \times 100\%$ of the data from $A^c$. Instances segmentation is then performed on the union $A \cup B$ according to the variance gain $\widetilde{V}_j(d)$:

$$\widetilde{V}_j(d) = \frac{1}{n} \left( \frac{\left( \sum_{x_i \in A_l} g_i + \frac{1-a}{b} \sum_{x_i \in B_l} g_i \right)^2}{n_l^j(d)} + \frac{\left( \sum_{x_i \in A_r} g_i + \frac{1-a}{b} \sum_{x_i \in B_r} g_i \right)^2}{n_r^j(d)} \right). \tag{10}$$

Among them, the coefficient $(1-a)/b$ is used to increase the weight of small gradient samples in the subset, with $A_l$ and $A_r$ representing the left and right subsets of subset $A$ divided by the partition point, and $B_l$ and $B_r$ corresponding to the left and right subsets of subset $B$. Different from the traditional ensemble learning model, LightGBM dramatically reduces the computational cost by calculating $\widetilde{V}_j(d)$ in a small instance subset.

The PSO algorithm is a reliable and effective optimization algorithm with few parameters, rapid convergence speed, and straightforward operation. Integrating the PSO algorithm into the LightGBM framework enables the adaptive search for optimal parameters, such as the learning_rate, max_depth, min_data_in_leaf, and feature_fraction.

We randomly initialize a group of particles with an initial speed $v_0$ and position $x_0$, and define the fitness function. The speed $v_{id}$ and position $x_{id}$ of the $i$th particle in $d$-dimensional space are then iteratively updated, and the optimal position $p_{id,\text{pbest}}$ and $p_{id,\text{gbest}}$ are concurrently updated:

$$\begin{cases} v_{id}^{k+1} = \omega v_{id}^k + c_1 r_1 \left( p_{id,\text{pbest}}^k - x_{id}^k \right) + c_2 r_2 \left( p_{id,\text{gbest}}^k - x_{id}^k \right) \\ x_{id}^{k+1} = x_{id}^k + v_{id}^{k+1} \end{cases} \tag{11}$$

where $\omega$ is an inertia weight to balance the global convergence and convergence speed, $k$ represents the current number of iterations, $c_1$ and $c_2$ are positive learning factors, and $r_1$ and $r_2$ are two random digits generated in the range of [0, 1]. The procedure terminates once the maximum iterations are met or the difference between the optimal solutions of the two iterations is below the threshold.

### 2.2.7. Validation

To evaluate the efficacy of the new SDB method, the bathymetry is validated using the determination coefficient ($r^2$), root mean square error (RMSE), and mean absolute error (MAE):

$$r^2 = 1 - \frac{\sum\left(y_{\text{image}} - y_{\text{field}}\right)^2}{\sum\left(y_{\text{field}} - \bar{y}_{\text{field}}\right)^2} \tag{12}$$

$$RMSE = \sqrt{\frac{\sum (y_{\text{image}} - y_{\text{field}})^2}{N}} \quad (13)$$

$$MAE = \frac{\sum |y_{\text{image}} - y_{\text{field}}|}{N} \quad (14)$$

where $y_{\text{field}}$ is the in situ measured depth, $y_{\text{image}}$ is the derived depth, $\overline{y}_{\text{field}}$ is the mean of $y_{\text{field}}$, and $N$ is the size of the validation dataset. The $r^2$ is widely applied to evaluate the consistency between the predicted and actual values in regression models, and the closer $r^2$ is to 1, the greater is the model's explanatory ability of factors influencing depth.

## 3. Results

### 3.1. Bottom Reflectance Inversion and Benthic Habitat Mapping

To generate a generic bottom reflectance for the study site, Ma's bottom reflectance inversion approach is used [47]. Figure 5a shows the pseudo-color image with range [0, 255] generated by the bottom reflectance, which indicates that bottom reflectance changes with offshore distance, thus suggesting different substrate types across the study area. After graying the bottom reflectance image and inputting it into the improved ADFLICM algorithm proposed in Section 2.2.5, a substrate segmentation map (Figure 5b) can be generated in the absence of field measurements.

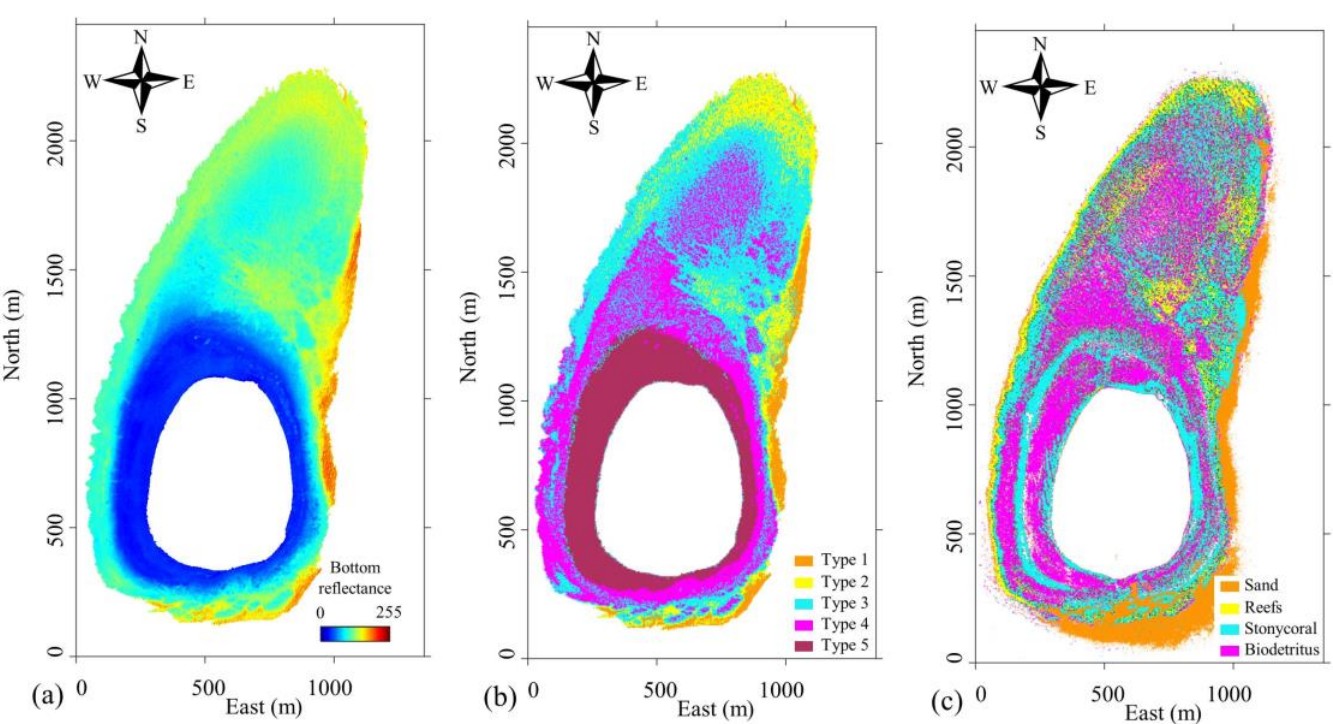

**Figure 5.** Bottom reflectance and substrate segmentation results over the research region. (**a**) Scaled seafloor reflectivity image, (**b**) segmentation result of substrate type derived from the proposed model, and (**c**) results of the substrate type distribution survey conducted in the area.

The new algorithm creates prior information that is independent and parameter adaptive, providing advantages over typical fuzzy c-means algorithms (FCMs). The results of the bottom-type clustering are in agreement with the coral reef habitat mapping products generated using field investigation, as shown in Figure 5c. A set of 100 sampling points in the experimental area were randomly selected for statistical consistency relative to the actual substrate profile (Table 2). Figure 5 and Table 2 demonstrate the consistency between the substrate type clustering results based on bottom reflectance and the actual substrate

distribution. Regions with relatively low reflectance have been independently classified as type 5. In reality, both bioclastic and stony coral are present in the coverage of type 5, with bioclastic being the more likely representative in this area. Table 2 confirms that the overall fit degree (the ratio of the fit category sample to the total sample) for seafloor substrate segmentation based on bottom reflectance without prior knowledge is 72%.

**Table 2.** Statistics of consistency between clustering results and actual substrate type distribution.

| Category | Sand | Reefs | Stony Coral | Biodetritus | Fit Category |
|---|---|---|---|---|---|
| Type 1 | 0.857 | 0.143 | 0.000 | 0.000 | Sand |
| Type 2 | 0.000 | 0.700 | 0.200 | 0.100 | Reefs |
| Type 3 | 0.039 | 0.115 | 0.769 | 0.077 | Stony coral |
| Type 4 | 0.062 | 0.094 | 0.156 | 0.688 | Biodetritus |
| Type 5 | 0.000 | 0.000 | 0.400 | 0.600 | Biodetritus |

*3.2. Water Depth Factor*

Corresponding to Table 1, the extraction of water depth factors in the study area covering water turbidity parameters, and the reflectance recorded by different wavelengths as well as their combinations has been completed. As shown in Figure 6, each parameter is uniformly scaled to the range of 0 to 255 for pseudo-color display, with the label being consistent with the numbers in Table 1, and the color bar interval being adjusted to emphasize the details.

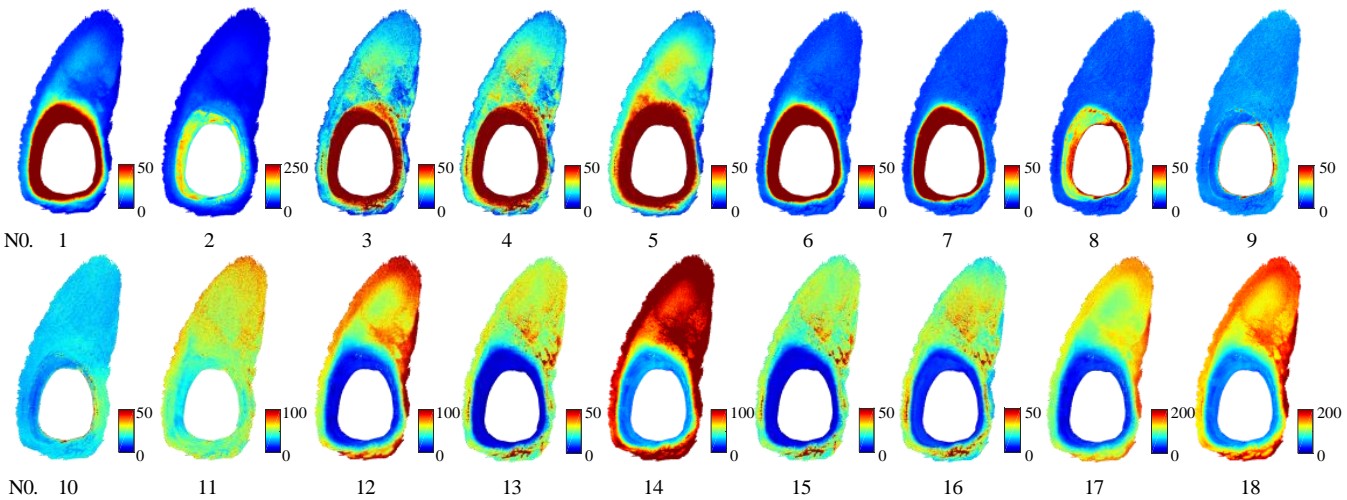

**Figure 6.** Extracted water depth factors over the entire study area.

In this work, a strategy for constructing corresponding water depth regression models for different substrate types was proposed to reduce the bottom type interference in the process of SDB. Various water depth factors had different abilities to map water depth in theory, so it was necessary to evaluate their importance before inputting them into the regression model for training. After the bottom reflectivity clustering, the correlation coefficients between the water depth factors and the measured water depth for each bottom type were assessed, as shown in Figure 7.

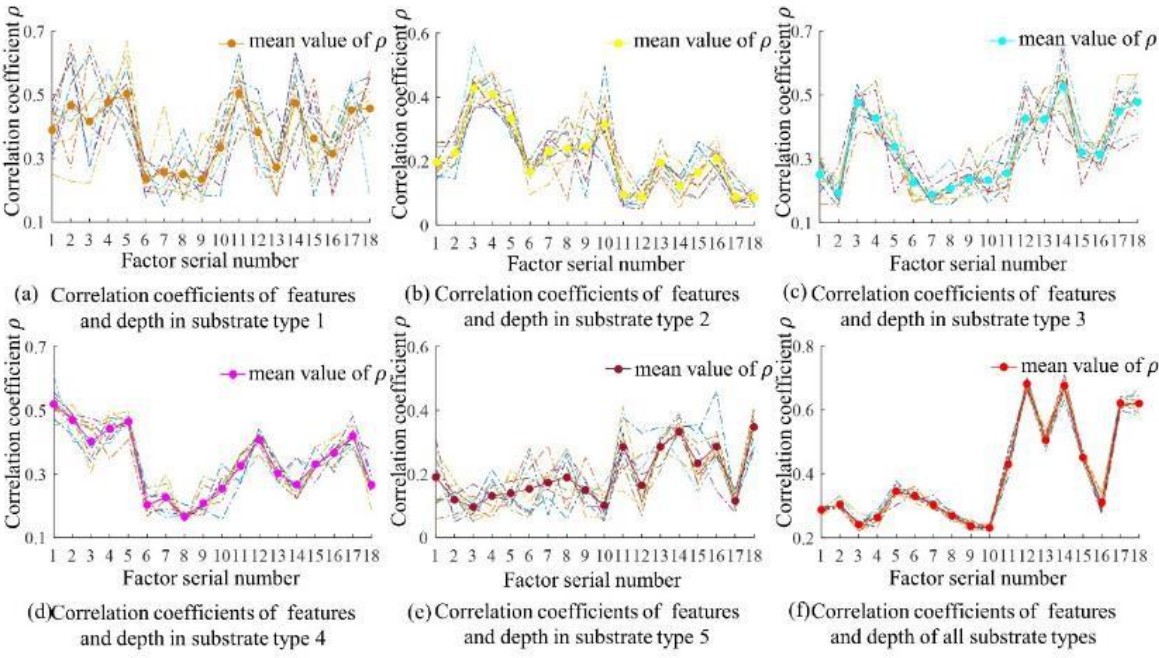

**Figure 7.** Correlation coefficients between characteristics and water depth under different substrate types. Dotted lines of different colors represent the correlation coefficients between different water depth factors in a random sampling operation.

Randomly selecting 400 water depth samples corresponding to each type of substrate, the correlation coefficients between different water depth factors were counted and the operation was repeated 10 times to minimize the interference of accidental errors. The water depth factor correlation analysis results from bottom type 1 to 5 are represented by (a) to (e) in Figure 7, respectively, with the mean of 10 outputs being plotted in bold. It can be seen that for substrate type 4, suspended sediment and chlorophyll concentrations, with the serial number 1 and 2 respectively, are the first two factors with the highest correlation. For substrate type 1 and 2, the above two factors are also among the ten factors with strong correlation. We further evaluated the correlation coefficient across the entire study area, without considering the bottom type, by randomly selecting 600 sample data and repeating the process ten times (Figure 7f). It was evident that the order of correlation of water depth factors varied between different bottom types. Some factors were not affected by the substrate type and were suitable for global inversion, while the performance of other factors varied significantly between different substrates.

### 3.3. Estimate Bathymetric Maps with In Situ Depth Points

Calculations based on correlation coefficients give feature importance ranking for derived depth, where the top ten features are input into a PSO-LightGBM regression model for model training. The adaptive search for model parameters using the PSO algorithm and bottom type classification with bottom reflectance provides the LightGBM method with a distinct advantage in terms of regression accuracy. The bathymetric map of the entire study area (Figure 8a) is generated based on the BA-SDB proposed in this paper. A set of 4000 sample data distributed as shown in Figure 8b, of which 2000 are used for model training and the remaining for model testing, are randomly selected to evaluate the water depth derivation. A set of 4000 sample data distributed as shown in Figure 8b, of which 2000 are used for model training and the remaining for model testing, are randomly selected to evaluate the water depth derivation accuracy of the BA-SDB model. The processing includes adaptive clustering of substrate types and feature evaluation before inputting the sample feature matrix and the corresponding actual water depth into the BA-SDB model. This is followed by parameter optimization, model training, and testing. Figure 8c

illustrates the error analysis of the derived water depth generated by the BA-SDB model with an $r^2$ value of 0.94, RMSE value of 0.85 m, and MAE value of 0.60 m.

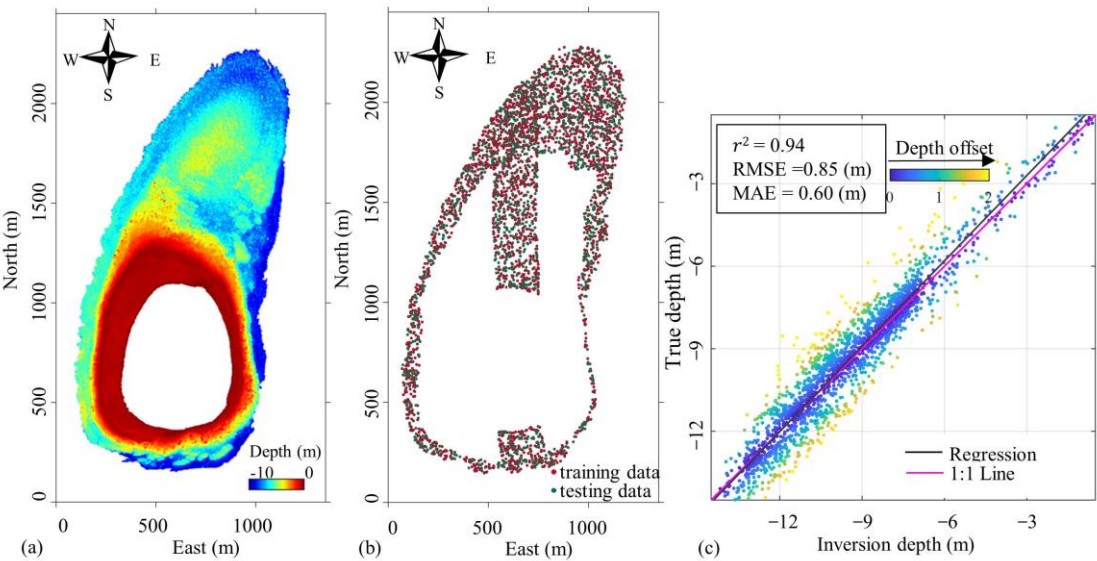

**Figure 8.** Comparison of derived water depth and measured bathymetry. (**a**) The derived bathymetry, (**b**) the distribution of the samples, and (**c**) the error analysis of the derived water depth.

## 4. Discussion

### 4.1. Assessment of Substrate Clustering

The clustering parameters should be consistent with the total number of substrate types in the study area given knowledge of the substrates. How to complete the bottom type classification without prior information confuses the subsequent processing. In this paper, the Calinski–Harabasz index is introduced into the ADFLICM approach to evaluate the clustering results and determine the clustering parameters adaptively. We tried to search the best clustering parameters from 3 to 7, and obtained the corresponding substrate segmentation map as shown in Figure 9. Visually, the substrate type is refined with the increased number of clustering parameters. However, when the clustering parameters exceed the actual substrate types, no definite substrate will correspond to some of the categories.

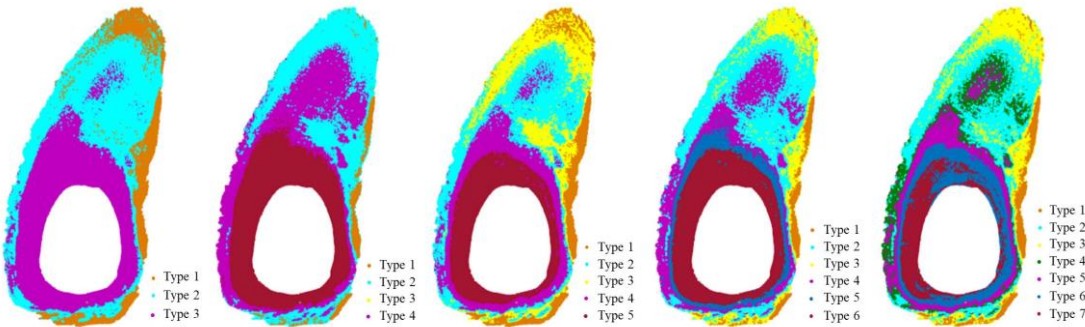

**Figure 9.** Bottom-type maps generated from different clustering parameters.

Table 3 provides statistics on the bottom type segmentation accuracy of 200 samples selected randomly employing the Calinski–Harabasz index and fit degree for different clustering parameters. The Calinski–Harabasz index (CHi) and fit degree showed a small peak at 5 clusters, and reached the peak at 7 clusters. However, the final clustering parameter is locked to 5, which balances the evaluation index and the actual substrate type, and the fit degree tends to be stable at 72%, and the CHi reaches the relative high value of 1354.

**Table 3.** Segmentation precision of bottom type for different clusters.

| Parameter/Evaluation Index Name | Value | | | | |
|---|---|---|---|---|---|
| Clusters | 3 | 4 | 5 | 6 | 7 |
| CHi | 368 | 641 | 1354 | 1317 | 1401 |
| Fit degree | 0.32 | 0.44 | 0.72 | 0.67 | 0.74 |

### 4.2. Evaluation of Bottom Type Clustering in Bathymetric Derivation

To further verify the effect of substrate division on water depth derivation, the water depth derivation accuracy of BA-SDB, which combines PSO-LightGBM algorithm and bottom type clustering, and the PSO-LightGBM model are designed and evaluated. The difference between the two methods is in the emphasis placed on the bottom type, where the former integrates bottom type classification and corresponding water depth factor screening based on PSO-LightGBM. Figure 10 shows the deviation between the derived water depth and the actual water depth tested on 100 samples randomly selected from each bottom type. Comparing the derived water depth obtained by the two methods, it was confirmed that the bottom category division plays a significant role in improving the accuracy of water depth derivation. Constructing regression models for different substrate types can better express the mapping relationship between water depth factors and water depth, and obtain accurate underwater terrain products. This is consistent with the results obtained, where the RMSE of the BA-SDB model in all bottom types is significantly lower than that of the PSO-LightGBM model.

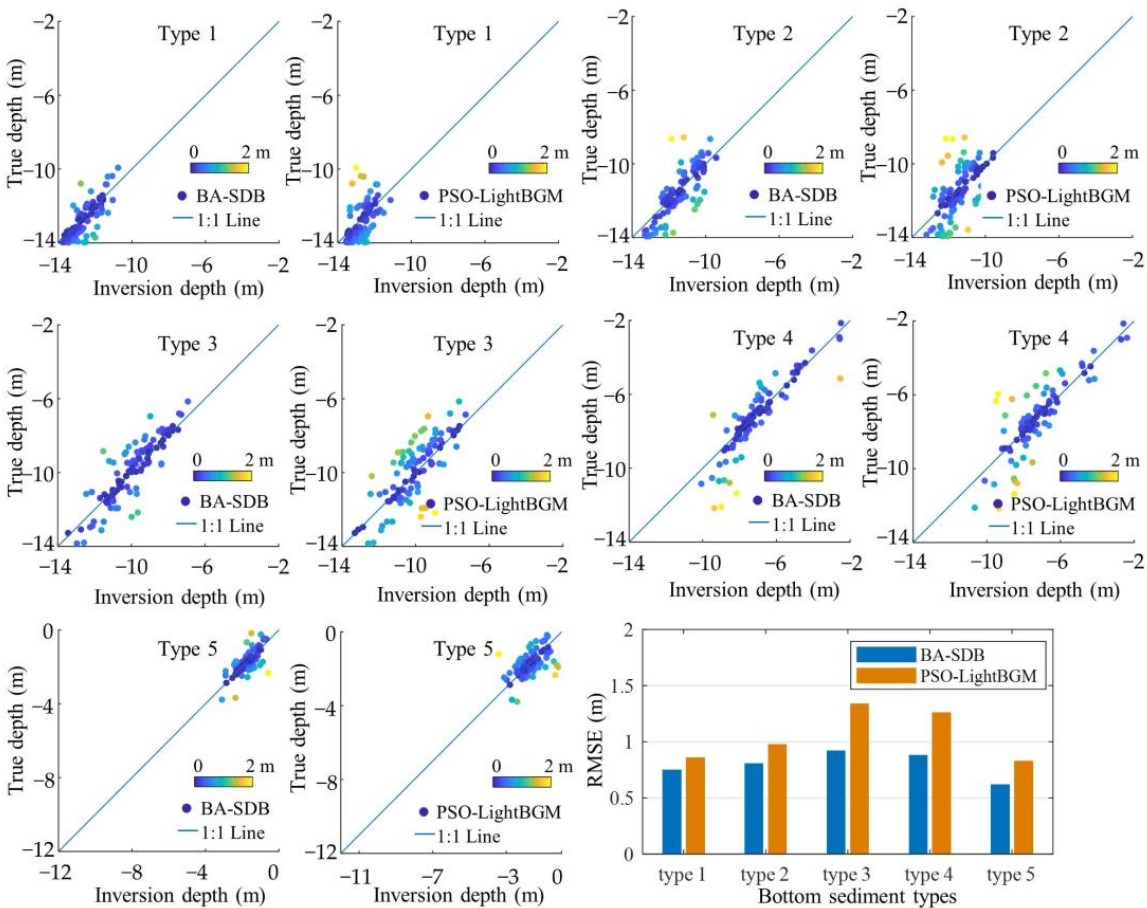

**Figure 10.** Accuracy evaluation of water depth derived from different substrate types. For each type, the result generated by the BA-SDB method is compared with that generated by the PSO-LightGBM method, and the dots in different colors represent the extent of depth offset. The histogram in the lower right corner counts the error analysis results of all substrate types for the two methods.

However, problems that come with substrate division need to be faced as well. As can be seen from Figures 5 and 8, there are inconsistencies in the water depth values at the boundaries of different substrate types. For example, the boundary between substrate type 1 and 3 on the south side is also the irregular line where water depth changes sharply. The outliers in the northwest shoal of the island correspond to substrate type 3 scattered in type 4. There may be some steps that can be taken to smooth the boundary and the adjacent pixels, but currently we have not addressed the inconsistencies. This is the key point that we need to pay further attention to.

### 4.3. Validity of the Depth Derivation Model

As the basic regression model of BA-SDB, the LightGBM framework is challenged by its multi-parameter setting, so the PSO algorithm is utilized to search for the optimal model parameters. The key parameters are initialized according to Table 4 and the default settings are enabled for the parameters not considered. One thousand samples were randomly selected to calculate the bathymetry derivation accuracy before and after the model parameter optimization for a quantitative analysis of the optimization effect. The iterative calculation to determine the optimal configuration of the approach employed the mean relative error as the evaluation index and resulted in the selection of the primary parameters for LightGBM: "learning_rate" is set to 0.05, "max_depth" is 5, "min_data_in_leaf" is 26, and "feature_fraction" is 0.6. Table 4 presents the changes in RMSE under different parameter settings. The PSO-LightGBM prediction is characterized by a lower RMSE value, indicating that parameter optimization is beneficial for improving the prediction accuracy of the bathymetry-derived regression model. An appropriate learning rate can theoretically lead to a stable and excellent model, while max_depth helps to prevent overfitting and improves the model's generalization ability, as does min_data_in_leaf. Feature_fraction is related to the training speed.

**Table 4.** Comparison of LightGBM parameters before and after optimization and the associated water depth accuracy.

| Parameter | Learning_Rate | Max_Depth | Min_Data_in_Leaf | Feature_Fraction | RMSE (m) |
|---|---|---|---|---|---|
| Before optimization | 0.1 | 10 | 20 | 0.8 | 1.31 |
| After optimization | 0.05 | 5 | 26 | 0.6 | 1.16 |

Figure 11 compares the predicted bathymetry, obtained from various approaches such as the Stumpf model, multiple linear regression model with reflectance of blue, green, and blue/green, random forest (RF), LightGBM, PSO-LightGBM, and BA-SDB, with the observed water depth. A set of 2000 samples is randomly selected and input into each regression model for accuracy index calculation. The input matrix of RF, LightGBM, and PSO-LightGBM is composed of the top 10 features in Figure 7f. Figure 11 displays that the BA-SDB method has the highest accuracy in predicting the sample data in the research area, while PSO-LightGBM, LightGBM, and RF follow in terms of accuracy. The results in Figure 11 confirm that the LightGBM model has superior regression performance over band ratio, polynomial fitting, and the RF model. Furthermore, parameter optimization and substrate type division have been shown to be effective in improving the accuracy of water depth estimation.

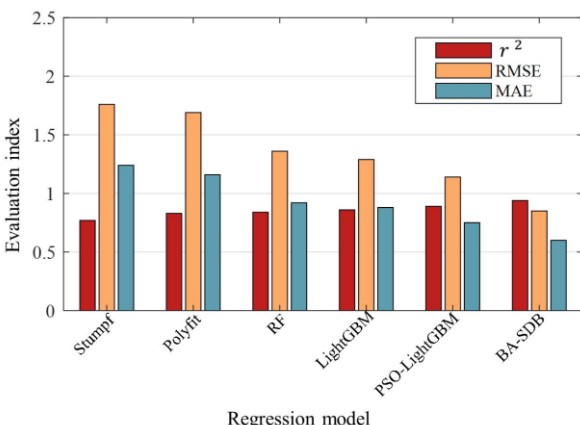

**Figure 11.** Comparison of bathymetry derivative accuracy of different regression models.

For the Stumpf model, it is impossible to make full use of all the bands provided by multispectral imagery. While multiple linear regression models can utilize all bands, the Stumpf model is subject to its relatively fixed mode and little consideration of inhomogeneous water environments. Therefore, the machine learning models with variable structure and multiple inputs always perform better than the conventional models. However, it should not be ignored that although the proposed model realizes adaptive partition and requires little manual intervention, it takes time for substrate segmentation and factor selection, so the efficiency is inevitably lower than that of the conventional model.

## 5. Conclusions

Coastal bathymetry is vital for various fields, including marine transportation, marine science research, and coastal zone planning and management. With rising sea levels and changing coastal dynamics, the need for reliable coastal bathymetry data is even more critical now than ever before. There is an urgent need to develop a more adaptive and accurate SDB method to bridge the gap between data requirements, cost, and the ability to map through shipborne- or airborne-based sensors. Previous studies on SDB aimed to reduce the impact of environmental factors and improve the quantitative level of bathymetry remote sensing interpretation. However, the distinct distribution of water turbidity and substrate in different water areas can result in different radiation and thus limit the accuracy of water depth remote sensing. Achieving self-adaptability and derivation accuracy is essential for SDB model migration and application. To this end, we strive to achieve a balance between adaptability, cost, time, and accuracy and provide a highly automated and accurate SDB method.

This study presents an adaptive and empirical SDB approach based on bottom type. By constructing a nonlinear regression for each substrate type, this method eliminates the adverse effects of substrate differences on SDB and can also screen the advantageous characteristic matrix. Detailed comparative experiments and quantitative analysis demonstrate the effectiveness and reliability of this new method.

Research has shown that water turbidity has a significant impact on the accuracy of optically derived bathymetry from multispectral imagery. Different levels of turbidity scatter incoming radiation differently, resulting in varying effects on the accuracy of optically sensed water depth. The backscattering of incident light from water increases with the increase in SSC level, resulting in higher pixel values. To address the limitation of the conventional SDB model, which only considers reflectivity, the introduction of water environment parameters in the inversion model is necessary to improve accuracy. Therefore, the introduction of water environment parameters in the regression model makes up for the confines of the conventional SDB model, which only considers reflectance. In this research, water turbidity factors were used as general water depth factors in the mapping relationship with depth, which could limit their effectiveness in SDB. Therefore, in

future studies, water turbidity should be given as much attention as the bottom type when deriving optical depth.

**Author Contributions:** Conceptualization, X.J. and Y.M.; methodology, X.J.; software, X.J.; validation, X.J., J.Z. and W.X.; formal analysis, X.J.; investigation, Y.W.; resources, X.J.; data curation, X.J.; writing—original draft preparation, X.J.; writing—review and editing, J.Z., W.X. and Y.W.; visualization, X.J.; supervision, X.J.; project administration, Y.M.; funding acquisition, Y.M. All authors have read and agreed to the published version of the manuscript.

**Funding:** This research was funded by the open fund of Technology Innovation Center for Ocean Telemetry, Ministry of Natural Resources (grant number 2022002), the open research fund program of LIESMARS (grant number 22S02), the National Natural Science Foundation of China (grant numbers 41871381, 42171407, 42077242), and the Natural Science Foundation of Jilin Province (grant number 2021010109 8JC).

**Data Availability Statement:** Not applicable.

**Acknowledgments:** The authors would like to sincerely thank the editors and the anonymous reviewers.

**Conflicts of Interest:** The authors declare no conflict of interest.

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
