# Peer review of "A Sub-Bottom Type Adaption-Based Empirical Approach for Coastal Bathymetry Mapping Using Multispectral Satellite Imagery"

_remotesensing, doi:10.3390/rs15143570_

Round 1

Reviewer 1 Report

The authors submitted a well written and an interesting manuscript dealing satellite derived bathymetry in shallow water using multi-spectral satellite images. Given the current changes in climate system and its effects in coastal areas, the current study contributes to scientific effort in searching for tools needed to monitor and mitigate these effects on coastal ecosystem. However, the manuscript needs to be revised before it could be considered for publication. Bellow are some comments and suggestions to improve the overall quality of this manuscript:

Lines 25-27: Please rephrase this statement “The results of this study confirm the effectiveness of the water turbidity and substrate in a machine learning method for bathymetric mapping” because it is not clear.

Line 105: Please rename the Section 2 from 2. Study site and data to 2. Materials and Methods and start it with subsection containing the descriptions of the Study Area and follow it by another subsection of Methodology in which you can include the descriptions of the datasets and the methods used in this study.

Lines 121-123: The description of the in-situ data provided is not sufficient. Please provide their accuracy and precision and their file format as they are provided. Please provide some description on their acquisition.

Lines 137-138: The Figure 2 is not clear, please provide a sharp image, if the copy right is needed, please provide it, indicate the reference.

Lines 145, 161: The authors provided two subsections with the same Title, please check, and rectify.

Line 300: Please provide the unity for the legend of the Figure 6.

Lines 53-54: Please provide references of recent works conducted using remote sensing technology tools for mapping coastal shallow water bathymetry. Please refer to (1) Muzirafuti, A.; Crupi, A.; Lanza, S.; Barreca, G.; Randazzo, G. Shallow water bathymetry by satellite image: A case study on the coast of San Vito Lo Capo Peninsula, Northwestern Sicily, Italy. In Proceedings of the IMEKO TC-19 International Workshop on Metrology for the Sea, Genoa, Italy, 3–5 October 2019; (2) Zhang, X.; Chen, Y.; Le, Y.; Zhang, D.; Yan, Q.; Dong, Y.; Han, W.; Wang, L. Nearshore Bathymetry Based on ICESat-2 and Multispectral Images: Comparison between Sentinel-2, Landsat-8, and Testing Gaofen-2. IEEE J. Sel. Top. Appl. Earth Obs. Remote Sens. 2022, 15, 2449–2462.

Minor editing of English language required

Reviewer 2 Report

Comments on the paper: “A sub-bottom type adaption-based empirical approach for coastal bathymetry mapping using multispectral satellite imagery” 

This study proposes a bottom-type adaption-based Satellite-derived bathymetry approach (BA-SDB). Under the consideration of multiple factors including suspended substrate and chlorophyll-a concentration, it uses particle swarm optimization improved LightGBM algorithm (PSO-LightGBM) to derive depth of each presegmented bottom type. Based on multispectral image of high spatial resolution and in-situ observations of airborne laser bathymetry and multi-beam echo sounder, the proposed approach is applied in shallow water around Yuanzhi island, and achieves the highest accuracy with an RMSE value of 0.85 m comparing to log-ratio, multi-band and classical machine learning methods.

This paper is a very interesting and meaningful paper! The exploration is innovative and the findings insightful, therefore it is worth publication. However, there is a lack of explanation in many research details and this paper could be improved in the following aspects.

1. It can be seen from Figure 1.(b) that in the measurement area where the multi-beam sonar and ALB overlap, the water depth values are not consistent, and there are still big differences. Please explain how to deal with it?

2. The new model takes into account factors such as suspended sediment and chlorophyll concentration, but it does not explain which ten factors with strong correlation are selected in the final selected water depth factor, and whether these factors are related to suspended sediment and chlorophyll concentrations?

3. The water depth results obtained based on different sub-bottom type may have inconsistencies in the water depth values at the boundaries of different sub-bottom types. It is recommended that the author conduct corresponding analysis work.

4. Figure 10 lacks a legend for depth offset. Please correct it.

5. The titles of sections 3.2 and 3.3 are duplicated.

/

Reviewer 3 Report

Thank you. Please check the attached file for feedback and comments. Nice works

Reviewer 4 Report

The manuscript titled "A sub-bottom type adaption-based empirical approach for coastal bathymetry mapping using multispectral satellite imagery" introduces a novel empirical method for deriving bathymetry information from multispectral satellite imagery. The proposed approach is designed to account for various types of substrates found in coastal regions. The manuscript is well-structured, well-written, and falls under the domain of "Remote Sensing," making it a relevant contribution to the field.

Major comments

While the model's description is clear, there is space for further enhancement by elaborating on the specific data requirements needed to apply the model to different areas. In fact, it is vey important to describe the data needs for utilizing the model effectively in other areas. The comparison of various models should also encompass a discussion of these data requirements, as they play a crucial role in the application of each model. Furthermore, discussing the data requirements in the context of model comparison would provide valuable insights into the trade-offs associated with each approach. Researchers could consider factors such as data availability, data collection costs, and data quality. They could then assess whether a model with higher data requirements may offer superior performance or if a more modestly demanding model would suffice given the available resources.

In addition to the points mentioned earlier, the description should provide more clarity on the quality of the validation data. Specifically, the methods used to measure in-situ depth through multibeam echo sounder and airborne laser bathymetry need to be described in greater detail. These systems have varying performance characteristics, and it is important to provide information about their errors and spatial coverage. Additionally, it is known that laser-derived bathymetry can have significant errors, so it would be valuable to include specific information about the magnitude of these errors and how they were accounted for in the validation process.

Furthermore, the description of the "results of the substrate type distribution survey conducted" requires clarification. It would be helpful to explain the criteria used to classify the substrate as biodetritus and stonycoral. For instance, does biodetritus refer to biogenic sand, or is it a broader category encompassing other biogenic materials? Similarly, stonycoral is spelled as stony coral or Story coral, so consistent terminology should be used throughout the description.

Minor comments:

L20: what is a “suspended substrate”?

L215: eq 8 does not seam to represent Spearman cc.     

L260: the notation ?Ì‚field for the mean is not standard.

It would very useful to potential readers to have access to the code.   
